# Use of isotretinoin among girls and women of childbearing age and occurrence of isotretinoin-exposed pregnancies in Germany: A population-based study

Jonas Reinold[1], Bianca Kollhorst[2], Nadine Wentzell[1], Katharina Platzbecker[1], Ulrike Haug[1,3¤]*

1 Department of Clinical Epidemiology, Leibniz Institute for Prevention Research and Epidemiology – BIPS, Bremen, Germany, 2 Department of Biometry and Data Management, Leibniz Institute for Prevention Research and Epidemiology – BIPS, Bremen, Germany, 3 Faculty of Human and Health Sciences, University of Bremen, Bremen, Germany

¤ Current address: Leibniz Institute for Prevention Research and Epidemiology – BIPS, Bremen, Germany
* haug@leibniz-bips.de

## Abstract

### Background

Exposure to isotretinoin during pregnancy must be avoided due to its teratogenicity, but real-world data on its use are scarce. We aimed to describe (i) isotretinoin use in women of childbearing age in Germany; (ii) the occurrence of isotretinoin-exposed pregnancies; and (iii) malformations among children exposed in utero.

### Methods and findings

Using observational data from the German Pharmacoepidemiological Research Database (GePaRD, claims data from approximately 20% of the German population), we conducted annual cross-sectional analyses to determine age-standardized prevalence of isotretinoin use between 2004 and 2019 among girls and women aged 13 to 49 years. In cohort analyses, we estimated the number of exposed pregnancies by assessing whether there was prescription supply overlapping the beginning of pregnancy (estimated supply was varied in sensitivity analyses) or a dispensation within the first 8 weeks of pregnancy. Data of live-born children classified as exposed in a critical period according to these criteria were reviewed to assess the presence of congenital malformations.

The age-standardized prevalence of isotretinoin use per 1,000 girls and women increased from 1.20 (95% confidence interval [CI]: 1.16, 1.24) in 2004 to 1.96 (95% CI: 1.92, 2.01) in 2019. In the base case analysis, we identified 178 pregnancies exposed to isotretinoin, with the number per year doubling during the study period, and at least 45% of exposed pregnancies ended in an induced abortion. In sensitivity analyses, the number of exposed pregnancies ranged between 172 and 375. Among live-born children, 6 had major congenital malformations. The main limitation of this study was the lack of information on

**Data Availability Statement:** As we are not the owners of the data we are not legally entitled to grant access to the data of the German Pharmacoepidemiological Research Database. In accordance with German data protection regulations, access to the data is granted only to

employees of the Leibniz Institute for Prevention Research and Epidemiology – BIPS on the premises of the institute and in the context of approved research projects. Third parties may only access the data in cooperation with the Leibniz Institute for Prevention Research and Epidemiology – BIPS and after signing an agreement for guest researchers. Please contact gepard@leibniz-bips.de for help with this process.

**Funding:** The study was partly funded by the German Federal Institute for Drugs and Medical Devices, Bonn (BfArM, V-18281/ 68605 / 2019-2020) (https://www.bfarm.de). The study proposal was submitted by UH. The funders had no role in study design, data collection and analysis, decision to publish, or preparation of the manuscript.

**Competing interests:** The authors have declared that no competing interests exist. JR, NW, KP, BK, and UH are working at an independent, non-profit research institute, the Leibniz Institute for Prevention Research and Epidemiology – BIPS. Unrelated to this study, BIPS occasionally conducts studies financed by the pharmaceutical industry. These are post-authorization safety studies (PASS) requested by health authorities. The design and conduct of these studies as well as the interpretation and publication are not influenced by the pharmaceutical industry. The study presented was not funded by the pharmaceutical industry.

**Abbreviations:** ATC, Anatomical Therapeutic Chemical; CI, confidence interval; DDD, defined daily dose; FAERS, FDA Adverse Event Reporting System; GePaRD, German Pharmacoepidemiological Research Database; IQR, interquartile range.

the prescribed dose, i.e., the supply had to be estimated based on the dispensed amount of isotretinoin.

## Conclusions

Isotretinoin use among girls and women of childbearing age increased in Germany between 2004 and 2019, and there was a considerable number of pregnancies likely exposed to isotretinoin in a critical period. This highlights the importance of monitoring compliance with the existing risk minimization measures for isotretinoin in Germany.

## Author summary

### Why was this study done?

- Systemic (oral) isotretinoin is used in the treatment of moderate to severe acne.

- Given that isotretinoin is one of the strongest human teratogens known today, it is important to monitor the use of isotretinoin in girls and women of childbearing age as well as the occurrence of pregnancies exposed to this drug.

- There is a lack of population-based studies addressing these research questions.

### What did the researchers do and find?

- Using a database covering 20% of the German population, we conducted cross-sectional analyses to assess the prevalence of isotretinoin use between 2004 and 2019 in girls and women of childbearing age.

- We found that the age-standardized prevalence of isotretinoin use increased from 1.20 to 1.96 per 1,000 girls/women during this period.

- In cohort analyses, we estimated the number of pregnancies likely exposed to isotretinoin in a critical period. In the base case analysis, we identified 178 of such pregnancies.

- Sensitivity analyses considering the recommended one-month washout period suggested that there could have been additional pregnancies exposed to isotretinoin because they started before the end of the washout period.

### What do these findings mean?

- Isotretinoin use among girls and women of childbearing age increased in Germany between 2004 and 2019, and there were a considerable number of pregnancies likely exposed to isotretinoin in a critical period.

- This highlights the importance of monitoring compliance with the existing risk minimization measures for isotretinoin in Germany.

- It also seems important to increase awareness regarding the component of the pregnancy prevention program that recommends contraception also in the month after treatment cessation.

- The main limitation of this study was the lack of information on the prescribed dose of isotretinoin. We therefore estimated the dose based on the dispensed amount of isotretinoin and varied the underlying assumptions.

## Introduction

Systemic (oral) treatment with the vitamin A derivative isotretinoin (13-cis-retinoic acid) is indicated in moderate to severe acne (e.g., nodular or conglobate acne or acne at risk of permanent scarring) resistant to therapy with systemic antibiotics and topical anti-acne treatment [1]. Systemic isotretinoin is considered to be the clinically most effective anti-acne therapy, achieving long-term remission or significant improvement in many patients [2,3]. At the same time, isotretinoin is one of the strongest human teratogens known today [4]. Given the age distribution of patients with acne, girls and women of childbearing age are among the patients to whom isotretinoin is prescribed.

In 2003, i.e., 20 years after the EU-market authorization for isotretinoin, uniform pregnancy prevention programs were established in order to avoid isotretinoin exposure during pregnancy. In 2018, these measures were evaluated and updated [5]. Girls and women initiating systemic isotretinoin treatment are required to have monthly pregnancy tests and to use 2 complementary contraceptive methods from 1 month before treatment initiation to 1 month after treatment cessation [5]. The continuation of contraception for 1 month after treatment cessation is recommended due to delayed plasma elimination [3]. Because of this delayed elimination, pregnancies beginning shortly after treatment with isotretinoin may also be exposed to potentially harmful plasma concentrations.

Given the potential harm to the unborn child, it is important to monitor the use of isotretinoin in girls and women of childbearing age as well as the frequency of pregnancies exposed to this drug. However, no such data are available from Germany. Also internationally, population-based studies quantifying the frequency of isotretinoin-exposed pregnancies are only available for few countries, namely the United States, Canada, France, and the Netherlands [6–11], and there are no studies on time trends of isotretinoin use among young women. Furthermore, there is a lack of studies systematically exploring the extent to which conceptions occurring in the month after discontinuation of treatment may contribute to the number of exposed pregnancies.

Therefore, the aims of this study were (i) to describe the utilization of isotretinoin in girls and women of childbearing age in Germany including time trends; (ii) to describe the frequency of pregnancies exposed to isotretinoin, considering also potential exposure due to treatment cessation in the month before pregnancy; and (iii) to explore potential malformations among children exposed to isotretinoin in early pregnancy (not to be mistaken with estimating causal effects).

## Methods

### Data source

We used the German Pharmacoepidemiological Research Database (GePaRD) which is based on claims data from 4 statutory health insurance providers in Germany and currently includes information on approximately 25 million persons who have been insured with one of the participating providers since 2004 or later. In addition to demographic data, GePaRD contains

information on drug dispensations as well as outpatient (i.e., from general practitioners and specialists) and inpatient services and diagnoses. The data is available on an individual level. Per data year, there is information on approximately 20% of the general population and all geographical regions of Germany are represented [12,13]. The German health care system is based on mandatory private or statutory health insurances [14]. About 90% of the general population are covered by statutory health insurances [15]. Core characteristics of the German health insurance system are uniform access to all levels of care and a free choice of providers.

In GePaRD, the Anatomical Therapeutic Chemical (ATC) code is used to identify drugs dispensed in the outpatient setting. Systemic isotretinoin treatment was identified based on the ATC code D10BA01. Diagnoses in GePaRD are coded according to the International Classification of Diseases 10th revision, German modification (ICD-10-GM). For research on drug utilization and safety during pregnancy, algorithms to identify and classify pregnancy outcomes [16,17], to estimate the beginning of pregnancy [18], and to link mothers with their children [19] have been developed for GePaRD. This study follows the Strengthening the Reporting of Observational studies in Epidemiology (STROBE) guidelines (S1 Checklist).

## Study design and study population

**Prevalent use of isotretinoin among girls and women of childbearing age.** To determine prevalent use of isotretinoin over time, we conducted year-wise cross-sectional analyses from 2004 to 2019. For each calendar year, we included all girls and women in the numerator who had at least 1 dispensation of isotretinoin, were aged between 13 and 49 years in the respective year, and were insured on June 30 of that year. In the denominator, we included all girls and women aged between 13 and 49 years in the respective year and insured on June 30 of that year.

**Identification of exposed pregnancies.** Using the algorithm for pregnancy outcomes, we identified pregnancies ending between 2004 and 2019 and occurring among girls and women aged 13 to 49 years at beginning of pregnancy.

Exposure to isotretinoin during early pregnancy was assumed if the exposure window assigned to the last dispensation before pregnancy overlapped the first day of pregnancy or if there was a dispensation in the first 8 weeks of pregnancy. The latter time period was restricted to 8 weeks rather than 12 weeks because it is then more likely that the dispensed drug was actually used during the first trimester, so this was a conservative approach to avoid overestimating the number of pregnancies exposed during the first trimester. The exposure window assigned to the last dispensation before pregnancy was defined as the dispensation date plus the total number of defined daily doses (DDDs) in the package (1 DDD of isotretinoin is 30 mg). In various sensitivity analyses, we changed the exposure window to take into account (i) delayed elimination (30 days were added to the exposure assigned to the last dispensation); (ii) potential treatment at higher or lower doses (the DDD was multiplied by the factor 0.75, 1.5, and 3.0, respectively, which corresponds to a daily dose of 40 mg, 20 mg, and 10 mg, respectively); and (iii) delayed elimination in combination with treatment at higher or lower doses (the DDD was multiplied by the factor 0.75, 1.5, and 3.0, respectively, and in addition, a 30-day period was added). In addition (iv), we conducted a sensitivity analysis assuming a fixed supply of 30 days after the last dispensation before pregnancy as this is the maximum supply that should be dispensed to girls and women of childbearing age according to the German pregnancy prevention program for isotretinoin [20]. In each of the analyses, we made sure that the pre-observation period of the mothers included in the respective analysis before beginning of pregnancy was sufficiently long to assess the exposure status.

Given that certain incomplete pregnancies in claims data may have no outcome recorded (e.g., miscarriages not requiring medical treatment, induced abortions without medical

indication) and would therefore remain undetected when only applying the outcome algorithm, we also searched for this type of incomplete pregnancies. To qualify for this category, there had to be at least a code indicating the expected delivery date and another indicator of a pregnancy (e.g., a pregnancy-related examination) within a plausible time interval after the estimated beginning of pregnancy. The date of the last pregnancy-related examination recorded in the data was assigned as the end of these pregnancies. We determined the exposure status of these pregnancies as described above.

Pregnancies were followed up from the beginning to the end of pregnancy. Pregnancies that could not be followed up until the end due to an interruption in the mother's health insurance period, the end of the study period or the mother's death, the pregnancies were not excluded but we classified them as a separate category ("ongoing pregnancies") given that their outcome could not be determined within the available follow-up. When assessing the distribution of pregnancy outcomes, we only considered non-ongoing pregnancies.

**Exploration of potential malformations among exposed children.**   For exposed pregnancies ending in a live birth, we applied an algorithm linking mothers with their children [19] to explore potential congenital malformations in the children. Among linked children, we identified those with any malformation code (ICD-10-GM: Q00–Q99) occurring up to 1 year after birth. We reviewed all information available on them in GePaRD in order to verify the presence of malformations. The profile review was conducted independently by 2 reviewers (1 physician and 1 epidemiologist, both with expertise in the interpretation of codes in German claims data). Consensus was reached in a subsequent case conference. While reviewers were instructed to consider certain objective criteria confirming the occurrence of malformations, such as the presence of inpatient codes, repeated coding, and treatment or monitoring of the malformations, they were specifically asked to apply their clinical judgment in light of the overall patient history including, e.g., gestational age at birth or chromosomal abnormalities as potential alternative explanations for malformations. In doing so, the primary focus was on malformations categorized as "major" as suggested by EUROCAT, but so-called "minor" malformations were also considered if treatment (e.g., surgical) or other information (e.g., physical impairment, malformation-related complications) indicated a higher level of severity [21].

## Data analysis

In the cross-sectional analyses, we determined—for each year—age-specific and age-standardized prevalence of isotretinoin use. For age standardization, we used the age distribution of the German female population on 31 December 2019 as reference. Furthermore, we described the medical specialty of the prescribing physicians. For that purpose, we considered all isotretinoin dispensations in the respective year among included girls and women and assigned the specialty of the prescribing physician based on the information contained in the individual physician number [22]. As for pregnancies, we determined the number of those classified as exposed overall and by calendar year. We described the mothers' age at the beginning of pregnancy and the pregnancy outcomes.

The distribution of continuous variables was summarized as median (interquartile range [IQR]), while categorical variables were expressed as frequency counts (percentages). We conducted all statistical analyses using the software SAS version 9.4. The analyses were planned beforehand and conducted in an analogous way for other drugs with teratogenic potential [23]. There were no data-driven changes to the analysis plan. Two of the sensitivity analyses were conducted in response to reviewers (assuming a daily dose of 40 mg and a fixed supply of 30 days after the last dispensation before pregnancy, respectively).

### Ethics and approvals

In Germany, the utilization of health insurance data for scientific research is regulated by the Code of Social Law. All involved health insurance providers as well as the German Federal Office for Social Security and the Senator for Health, Women and Consumer Protection in Bremen as their responsible authorities approved the use of GePaRD data for this study. Informed consent for studies based on claims data is required by law unless obtaining consent appears unacceptable and would bias results, which was the case in this study. According to the Ethics Committee of the University of Bremen, studies based on GePaRD are exempt from institutional review board review. Specifically, for this study, the Ethics Committee of the University of Bremen was again asked whether a retrospective review might be needed. They judged that the information deducible from the publication is not suited—even in combination—to result in an identification of affected persons. They do thus not believe that there is sufficient risk to the individuals to warrant a retrospective review by an ethics committee. From a public health perspective, the information provided in this paper is very important to increase knowledge on the teratogenic effects of isotretinoin including the number and type of affected organs, all the more so as the use of isotretinoin is further increasing in women of childbearing age.

## Results

### Prevalence of isotretinoin use among girls and women of childbearing age

Overall, there were 50,936 girls and women with at least 1 prescription of isotretinoin in GePaRD between 2004 and 2019. Across all years, 52% to 67% of users were 16 to 30 years old, and 81% to 90% were ≤40 years old (S1 Table). The overall age-standardized prevalence of isotretinoin use per 1,000 girls and women increased from 1.20 (95% confidence interval [CI]: 1.16, 1.24) in 2004 to 1.96 (95% CI: 1.92, 2.01) in 2019, i.e., by 63% (Fig 1). The prevalences increased particularly among girls and women aged 13 to 30 years. For example, in age group 16 to 20 years they increased by 103% (from 2.07 [95% CI: 1.92, 2.23] to 4.20 [95% CI: 4.00, 4.41] per 1,000) and in age group 21 to 25 years they increased by 96% (from 1.96 [95% CI: 1.82, 2.11] to 3.84 [95% CI: 3.67, 4.02] per 1,000).

In total, 339,408 prescriptions of isotretinoin were dispensed to girls and women aged 13 to 49 years during the study period (S2 Table). The vast majority of those prescriptions (89%) was issued by dermatologists while general practitioners had a share of 6%.

### Exposed pregnancies

In the base case analysis, there were 178 pregnancies classified as exposed to isotretinoin during early pregnancy (Table 1). The number of exposed pregnancies per year increased over the course of the study period. On average, there were 7 pregnancies per year from 2004 to 2011 and 15 pregnancies per year from 2012 to 2019. The majority of the 178 pregnancies (62.9%) occurred in the age group 16 to 30 years (S3 Table).

The mothers' median age at the beginning of pregnancy was 28 years (IQR 24 to 33 years). About half of these pregnancies were classified as exposed due to a dispensation of isotretinoin within the first 8 weeks of pregnancy, about one quarter because the exposure window assigned to the last dispensation before pregnancy overlapped the beginning of pregnancy and in the remaining, both criteria were fulfilled.

For exposed pregnancies ending during the observation period ($n = 164$), the distribution of pregnancy outcomes is summarized in Table 1. Overall, 29.3% ($n = 48$) of exposed pregnancies ended in live birth, thereof 6.3% ($n = 3$) were preterm births, 45.1% ($n = 74$) ended in an

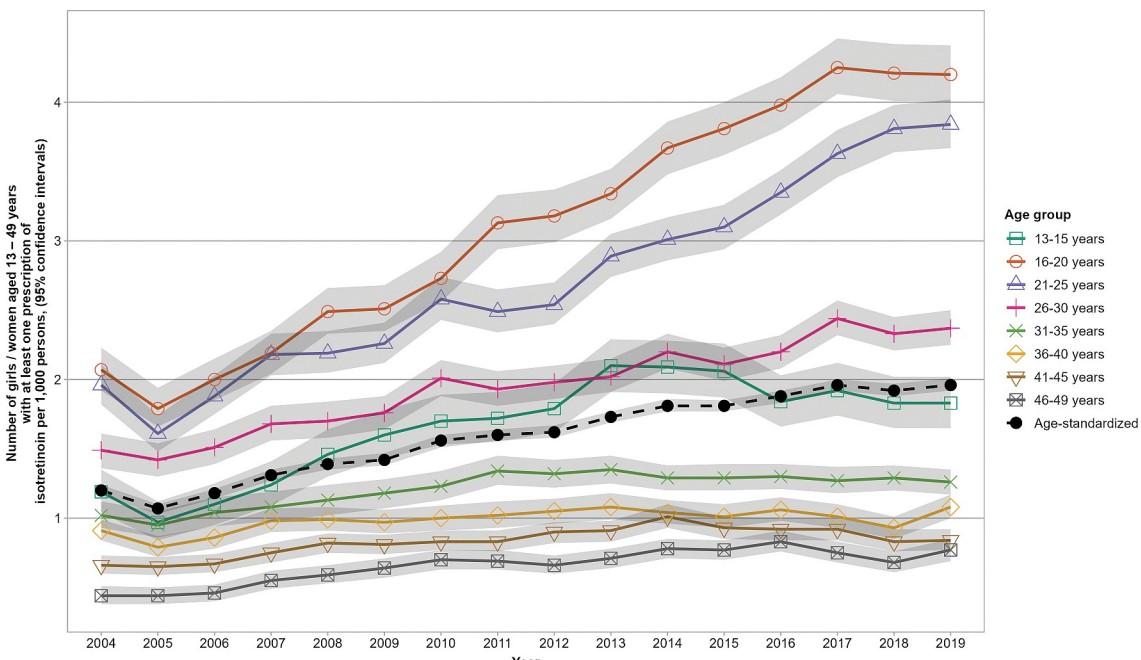

**Fig 1. Age-specific and age-standardized prevalence with 95% CIs (shaded area) of isotretinoin use per 1,000 girls and women aged 13–49 years between 2004 and 2019 in the GePaRD.** CI, confidence interval; GePaRD, German Pharmacoepidemiological Research Database.

induced abortion, and 1.8% (*n* = 3) in a miscarriage. For 20.7% (*n* = 34) of pregnancies, no outcome was recorded (i.e., they are assumed to also be abortions).

Table 1 also shows the number of pregnancies and the distribution of pregnancy outcomes observed in the sensitivity analyses. When delayed elimination (washout period of 1 month) was considered, the number of exposed pregnancies increased by 31% (from 178 to 233), while the proportion of live births and induced abortions remained at a level similar to the base case analysis. If a fixed supply of 30 days was assumed, there was an increase of exposed pregnancies by 7% (from 178 to 190). If a daily dose of 40 mg was assumed, the number of exposed pregnancies decreased by 3% (from 178 to 172), if a daily dose of 20 mg was assumed, it increased by 6% (from 178 to 188) and if a daily dose of 10 mg was assumed, it increased by 58% (from 178 to 283) compared to the base case analysis. When the one-month washout period was considered for assumed doses of 10 mg, 20 mg, and 40 mg, the increase in the number of exposed pregnancies was 33% (from 283 to 375), 39% (from 188 to 261), and 30% (from 172 to 224), respectively. In all sensitivity analyses, the proportion of live births was below 40% except for those with an assumed daily dose of 10 mg when delayed elimination was considered (46%).

## Characterization of exposed children

In 138 out of 157 (87.9%) pregnancies ending in a live birth and classified as exposed in the base case or the sensitivity analyses, the mother's and child's data could be linked. Of these, 6 children had at least 1 major congenital malformation according to EUROCAT definitions (Table 2). Five children had malformations classified as minor according to EUROCAT but are reported here as they required surgical or other intense treatment. Two other children had an atrial septal defect that could not be classified into major or minor according to EUROCAT given that the distinction requires information that is not available in German claims data.

**Table 1. Number of pregnancies exposed to isotretinoin between 2004 and 2019 in GePaRD, mother's age at pregnancy beginning and type of pregnancy outcome: base case and sensitivity analyses considering delayed elimination (exposure window assigned to the last dispensation before pregnancy was extended by 1 month), a fixed supply of 30 days, and the possibility that lower or higher doses than the DDD for isotretinoin (30 mg) were used.**

| | Base case analysis | Sensitivity analyses | | | | | | | |
|---|---|---|---|---|---|---|---|---|---|
| | Daily dose 30 mg (DDD) | (A) One-month extension | (B) Fixed supply of 30 days | (C) Daily dose 40 mg | (D) Daily dose 40 mg and one-month extension | (E) Daily dose 20 mg | (F) Daily dose 20 mg and one-month extension | (G) Daily dose 10 mg | (H) Daily dose 10 mg and one-month extension |
| **Number of exposed pregnancies** | 178 | 233 | 190 | 172 | 224 | 188 | 261 | 283 | 375 |
| Exposure overlapping pregnancy beginning, n (%) | 46 (25.8%) | 103 (44.2%) | 58 (30.5%) | 40 (23.3%) | 93 (41.5%) | 58 (30.9%) | 133 (50.9%) | 161 (56.9%) | 254 (67.7%) |
| Dispensation in the first 8 weeks of pregnancy, n (%) | 88 (49.4%) | 62 (26.6%) | 77 (40.5%) | 96 (55.8%) | 66 (29.5%) | 77 (40.9%) | 60 (23.0%) | 56 (19.8%) | 53 (14.1%) |
| Both, n (%) | 44 (24.7%) | 68 (29.2%) | 55 (28.9%) | 36 (20.9%) | 65 (29.0%) | 53 (28.2%) | 68 (26.1%) | 66 (23.3%) | 68 (18.1%) |
| **Mother's age** | | | | | | | | | |
| Median (Q1; Q3) | 28 (24; 33) | 28 (24; 33) | 28 (23; 32) | 28 (24; 32) | 28 (24; 33) | 28 (24; 33) | 28 (24; 33) | 28 (24; 33) | 28 (25; 33) |
| **Number of exposed and non-ongoing pregnancies[1]** | 164[1] | 213[1] | 175[1] | 158[1] | 205[1] | 172[1] | 240[1] | 259[1] | 340[1] |
| **Pregnancy outcomes** | | | | | | | | | |
| Live birth, n (%) | 48 (29.3%) | 63 (29.6%) | 52 (29.7%) | 44 (27.8%) | 61 (29.8%) | 49 (28.5%) | 81 (33.8%) | 103 (39.8%) | 157 (46.2%) |
| thereof preterm birth, n (% of live births) | 3 (6.3%) | 4 (6.3%) | 3 (5.8%) | 3 (6.8%) | 3 (4.9%) | 3 (6.1%) | 5 (6.2%) | 4 (3.9%) | 8 (5.1%) |
| Still birth, n (%) | 0 (0%) | 1 (0.5%) | 0 (0%) | 0 (0%) | 1 (0.5%) | 0 (0%) | 1 (0.4%) | 2 (0.8%) | 2 (0.6%) |
| Induced abortion, n (%) | 74 (45.1%) | 90 (42.3%) | 79 (45.1%) | 74 (46.8%) | 86 (42.0%) | 79 (45.9%) | 94 (39.2%) | 89 (34.4%) | 96 (28.2%) |
| Ectopic pregnancy or molar pregnancy, n (%) | 5 (3%) | 7 (3.3%) | 6 (3.4%) | 5 (3.2%) | 6 (2.9%) | 5 (2.9%) | 6 (2.5%) | 6 (2.3%) | 7 (2.1%) |
| Miscarriage, n (%) | 3 (1.8%) | 3 (1.4%) | 4 (2.3%) | 4 (2.5%) | 3 (1.5%) | 3 (1.7%) | 3 (1.3%) | 3 (1.2%) | 3 (0.9%) |
| No pregnancy outcome recorded[2], n (%) | 34 (20.7%) | 49 (23%) | 34 (19.4%) | 31 (19.6%) | 48 (23.4%) | 36 (20.9%) | 55 (22.9%) | 56 (21.6%) | 75 (22.1%) |

[1]This number is lower than the number of all exposed pregnancies in the respective analysis given that some pregnancies were still ongoing at the end of the observation period, so the outcome could not be determined yet. The percentages regarding the different types of pregnancy outcome refer to the number of non-ongoing pregnancies.

[2]There were clear indicators of a pregnancy but no outcome was recorded. It can be assumed that these pregnancies ended in a miscarriage not requiring medical care or an induced abortion not reimbursed by the health insurance.

DDD, defined daily dose; GePaRD, German Pharmacoepidemiological Research Database.

Of the 6 children with major malformations, 3 were classified as exposed in the base case analyses and 3 were classified as exposed in the sensitivity analyses. The major malformations affected the heart, the eye, the skull, and the nose. Furthermore, 1 child had sacral spina bifida with hydrocephalus and 1 child had multiple major malformations including microcephaly, malformation of the ear and heart. Among children with so-called minor malformations, 2 had congenital hypertrophic pyloric stenosis requiring surgical treatment, 2 had undescended testicles requiring surgical treatment and 1 had metatarsus varus with abnormalities of gait and mobility requiring extensive orthopedic treatment. Four of these 5 children were classified as exposed in the sensitivity analyses only.

Table 2 shows—for each of the 13 children with congenital malformations—in which of the analyses they were classified as exposed. In all of them, the exposure window assigned to the last dispensation before pregnancy overlapped the beginning of pregnancy. In the

**Table 2. Malformations observed in live-born children exposed to isotretinoin during pregnancy in GePaRD between 2004 and 2019.**

| Child | Q-Code | Description | Classified as "major" according to EUROCAT[2] | Base case | A | B | C | D | E | F | G | H |
|---|---|---|---|---|---|---|---|---|---|---|---|---|
| 1 | Q24.9 | Congenital malformation of heart, unspecified | Yes | X | X | X | X | X | X | X | X | X |
| 2 | Q40.0 | Congenital hypertrophic pyloric stenosis[3] | No | X | X | X | X | X | X | X | X | X |
| 3 | Q15.9 | Congenital malformation of eye, unspecified | Yes | X | X | X | X | X | X | X | X | X |
|  | Q10.3 | Other congenital malformations of eyelid | No |  |  |  |  |  |  |  |  |  |
| 4 | Q02 | Microcephaly | Yes | X | X | X | X | X | X | X | X | X |
|  | Q16.5 | Congenital malformation of inner ear | Yes |  |  |  |  |  |  |  |  |  |
|  | Q21.0 | Ventricular septal defect | Yes |  |  |  |  |  |  |  |  |  |
|  | Q25.6 | Stenosis of pulmonary artery[4] | Yes |  |  |  |  |  |  |  |  |  |
| 5 | Q40.0 | Congenital hypertrophic pyloric stenosis[3] | No |  |  |  |  |  |  | X | X | X |
| 6 | Q53.1 | Undescended testicle, unilateral[5] | No |  |  |  |  |  |  |  | X | X |
| 7 | Q21.1 | Atrial septal defect[6] | Unclear |  |  |  |  |  |  |  | X | X |
| 8 | Q75.0 | Craniosynostosis | Yes |  |  |  |  |  |  |  |  | X |
|  | Q01.0 | Frontal encephalocele | Yes |  |  |  |  |  |  |  |  |  |
| 9 | Q53.1 | Undescended testicle, unilateral[7] | No |  |  |  |  |  |  | X | X | X |
| 10 | Q75.8 | Other specified congenital malformations of skull and face bones | Yes |  |  |  |  |  |  |  | X | X |
|  | Q30.1 | Agenesis and underdevelopment of nose | Yes |  |  |  |  |  |  |  |  |  |
| 11 | Q66.2 | Metatarsus varus[8] | No |  |  |  |  |  |  |  | X | X |
| 12 | Q21.1 | Atrial septal defect[6] | Unclear |  |  |  |  |  |  |  |  | X |
| 13 | Q05.3 | Sacral spina bifida with hydrocephalus | Yes |  |  |  |  |  |  | X | X | X |

[1] In the base case analysis, a daily dose of 30 mg was assumed and the exposure window was not extended beyond the supply provided by the last dispensation before pregnancy. In the sensitivity analyses, the estimated daily dose was varied and/or the exposure window was extended to consider delayed plasma elimination. Specifically, the variations were as follows. A: one-month extension of the exposure window, B: a fixed exposure window of 30 days after the last dispensation was assumed, irrespective of the dispensed amount of isotretinoin, C: daily dose 40 mg, D: daily dose 40 mg and one-month extension of the exposure window, E: daily dose 20 mg, F: daily dose 20 mg and one-month extension of the exposure window, G: daily dose 10 mg, H: daily dose 10 mg and one-month extension of the exposure window.

[2] For those with minor malformation, information on treatment is provided to explain why they are reported here (see also Methods section).

[3] Hypertrophic pyloric stenosis was surgically treated.

[4] Classified as "major" as child was born on term (EUROCAT classification "minor" if gestational age at birth <37 week).

[5] Undescended testicle was surgically treated.

[6] The subtype of atrial septal defect is not reported in German claims data. "Unclear" in the column regarding EUROCAT classification means that although additional information coded for the child was considered, it was not clear whether the child had the subtype classified as minor according to EUROCAT (ICD-10 Q21.11) or one of the other subtypes classified as major. For child 7, ICD-10-GM Q21.1 was coded repeatedly (also in the inpatient setting) together with codes for cardiology visits and further diagnostics indicating monitoring beyond the first year of life. For child 12, ICD-10-GM Q21.1 was also coded repeatedly together with codes for cardiac murmur (ICD-10-GM R01), cardiology visits and further diagnostics indicating monitoring beyond the first year of life.

[7] Undescended testicle was surgically treated; in addition, there were codes for cardiac malformations (ICD-10-GM Q21.1, Q25.0, Q21.8) that are not listed in the table as the child was born prematurely.

[8] Child received extensive orthopedic treatment and had continuous records of ICD-10-GM R26.8: Other and unspecified abnormalities of gait and mobility.

GePaRD, German Pharmacoepidemiological Research Database.

children classified as exposed in the base case analysis, the mother additionally had a dispensation of isotretinoin within the first 8 weeks of pregnancy, while no such dispensations were observed in the mothers of children classified as exposed in the sensitivity analyses only.

## Discussion

In this population-based study covering approximately 20% of the German population, we found that the use of isotretinoin among girls and women of childbearing age increased by 63% between 2004 and 2019, from 1.20 per 1,000 in 2004 to 1.96 per 1,000 in 2019. This increase was particularly pronounced in girls and women up to the age of 30 years. Across the whole study period, more than 80% of users were ≤40 years (between 3,629 of 4,456 and 6,723 of 7,504), i.e., in age groups in which pregnancies typically occur. Even though risk minimization measures are in place, we observed—in the base case analysis—178 pregnancies likely exposed to isotretinoin during a time window most critical for fetal development. In sensitivity analyses varying the assumptions on the dose and considering the recommended one-month washout period, this number ranged between 172 and 375 pregnancies. In the base case analysis, at least 74 exposed pregnancies ended in an induced abortion (sensitivity analyses: 74 to 96). Among live births classified as exposed in the base case or the sensitivity analyses, there were 6 children with major malformations.

Regarding the increase in the prevalence of isotretinoin use among girls and women of childbearing age, there is hardly any data to which we could compare our findings. There is only a single data source reporting—for each year—the total number of DDDs dispensed to all persons with statutory health insurance in Germany (i.e., no denominator/prevalence estimate, and no age or sex-specific findings). Even though comparability is limited, that report supports our finding regarding an increase of oral use of isotretinoin (ATC code D10BA01) between 2004 and 2019 (2004: 5.2 million dispensed DDDs; 2019: 6.0 million dispensed DDDs) [24,25]. The mainstays of acne treatment have remained largely unchanged over recent years [26], so the increase in the prevalence of isotretinoin use cannot be explained by changes in treatment guidelines. Also, there have been no reports on major changes in the disease prevalence during the past 15 years. We can thus only speculate about reasons for this increase. German guidelines, which expired in 2014 without renewal, recommended systemic isotretinoin as second-line therapy after systemic antibiotics and topical anti-acne therapy, while the European guidelines recommend systemic isotretinoin as first-line therapy in moderate as well as severe acne [1]. In view of the effectiveness of systemic isotretinoin in the treatment of acne and the increasing awareness of the mental health burden associated with acne, German dermatologists might have been inclined to follow the European rather than the German guidelines. Another potential reason could be an increased awareness of antibiotic resistance related to the antibiotic stewardship programs launched during the past decade [27]. Considering recommendations from experts, regulatory authorities, and the WHO to shorten and reduce the number of treatment episodes with antibiotics, the threshold to proceed to second-line treatment with systemic isotretinoin might have become lower when treating acne patients [3,28].

In Germany, there is a long-established pregnancy prevention program for isotretinoin in line with EMA recommendations. In addition, prescriptions of isotretinoin must be filled within 7 days of being issued and the supply per prescription is limited to 30 days of treatment [20]. Despite these measures, we found—in the base case analysis—178 pregnancies likely exposed to isotretinoin in the critical time window. As our database covers approximately one fifth of the German population, it can roughly be estimated that at least 980 such pregnancies occurred across Germany between 2004 and 2019. As a matter of concern, the number of exposed pregnancies per year doubled during the study period, i.e., this problem is continuously gaining relevance. Our sensitivity analyses considering delayed plasma elimination and lower-dose treatment showed that the number of exposed pregnancies may even be considerably higher than estimated in the base case analysis. Of note, irrespective of the daily dose we assumed, the number of pregnancies classified as exposed increased by 30% to 39% if the one-

month washout period was considered. Since this suggests that pregnancies often occur too soon after the end of isotretinoin treatment, increasing awareness regarding the component of the pregnancy prevention program that recommends contraception also in the month after treatment cessation seems important.

Our study confirms reports from other countries about isotretinoin-exposed pregnancies occurring despite long-standing pregnancy prevention programs [29]. However, there is a lack of studies quantifying the frequency of these pregnancies. Dividing the number of exposed pregnancies by the total number of users in our study yields (roughly) a ratio between 3.5 (base case analysis) and 6.7 (sensitivity analyses) exposed pregnancies per 1,000 isotretinoin users. This is similar to the results of a study from the US based on pregnancy reports from the FDA Adverse Event Reporting System (FAERS) showing between 3.3 and 6.5 exposed pregnancies per 1,000 users between 1997 and 2017 [10]. It is also similar to a study based on Canadian administrative data (data from the provinces British Columbia, Saskatchewan, Manitoba, and Ontario) from 1996 to 2011, in which this ratio varied between 3.1 and 6.2 pregnancies per 1,000 isotretinoin users depending on the exposure definition (high-specificity versus high-sensitivity) [7]. Regarding Europe, population-based studies estimating the frequency of pregnancies per 1,000 users are only available for France where this ratio ranged between 0.32 and 0.95 when combining data from studies conducted between 1987 and 2011 [9]. A population-based study from the Netherlands determined the proportion of isotretinoin-exposed pregnancies among all pregnancies. It reported that between 1999 and 2007 about 2.5 per 10,000 pregnancies were exposed to isotretinoin in the 30 days before or during pregnancy [11].

The number of exposed pregnancies ending in a live birth in our study was 48 of 164 (29%) in the base case analysis and increased to 157 of 340 (46%) in a sensitivity analysis where a dose of only 10 mg and delayed elimination were considered. This illustrates that the proportion of live births is sensitive to the exposure definition, which may hamper comparability between existing studies. Furthermore, it is influenced by the extent to which incomplete pregnancies are captured in the respective database, which may also vary between studies. This may explain variation in the proportion of live births reported in different studies. For example, a German study using data collected in the context of counseling pregnant women and their health care providers reported a proportion of 18 live births of 91 pregnancies (20%) [30], a study from the United States reported proportions of 68 of 138 (49%) [8], while much lower proportions of 118 of 1,473 (8%) were reported by a Canadian study [7] and 85 of 553 (15%) in French studies [9].

While our study was not designed to quantify risks, it was still striking that 13 live-born children classified as exposed to isotretinoin in early pregnancy in the base case or the sensitivity analyses had congenital malformations, 6 of them with major malformations. Many of these malformations involved organ systems known to be affected by the so-called retinoic acid embryopathy. This finding and the fact that at least 45% of exposed pregnancies (74 of 164) in our study ended in an induced abortion—as compared to 4% when all pregnancies in GePaRD are considered [17]—underline the importance of improving adherence to pregnancy prevention programs.

Some limitations should be considered in the interpretation of our results. First, German claims data do not include the dose prescribed by the physician. Consequently, treatment duration has to be estimated based on the DDD. The DDD represents the dose for adults, but lower doses may be used, particularly in girls and adolescents, or sometimes a higher dose might be used. To address this issue, we performed comprehensive sensitivity analyses varying the exposure windows assigned to each dispensation. Second, as in all pharmacoepidemiological studies, there is uncertainty whether patients filling a prescription are actually taking the drug. It is also uncertain whether they always start taking the drug after filling the prescription

or whether they may partly start later. Third, while our study was designed to describe prevalence of isotretinoin use and pregnancies occurring under treatment with isotretinoin, our database would not have been suited to assess whether risk minimization measures were followed on an individual level. This would have required comprehensive information on contraceptive measures, which is limited in GePaRD as in most other claims databases [31]. Fourth, with regard to pregnancy outcomes and malformations, our study was merely descriptive, so causal conclusions cannot be drawn. Estimating causal effects would have required a different design including the consideration of relevant confounders, as well as a larger sample of exposed children. The latter might be achieved by a consortium of large databases. We strongly advise against simplified calculations in which the proportion of live births with malformations in our study is compared to the corresponding proportion reported for all live births in Germany in order to estimate risks. Such a comparison could be very misleading for various reasons (e.g., due to differences in age and thus presumably also in the prevalence of comorbidities, different proportions with induced abortions and thus missing malformation status). Fifth, as in most databases used to investigate drug utilization and safety during pregnancy, pregnancy outcomes clearly classified as "miscarriages" are underrepresented in our database, i.e., the frequency of this outcome was underestimated in our study. To address this limitation, we also searched for incomplete pregnancies with no outcome recorded. It seems plausible that these were miscarriages not requiring medical treatment or induced abortions without medical indication, i.e., not reimbursed by health insurances. Although not perfect as it still remained unclear whether it was a miscarriage or induced abortion, we think this approach was valuable to capture the number of exposed pregnancies more completely. Sixth, to assess the presence of malformations in children exposed during pregnancy, we conducted an in-depth patient profile review based on all diagnoses and procedure codes available in GePaRD but did not have additional clinical data.

A main strength of our study is the large claims database that has been shown to be representative of persons with statutory health insurance in Germany in terms of drug dispensations [32]. The available data allowed us to assess trends in isotretinoin dispensations over a 15-year period. Due to the use of claims data, our analyses were not affected by recall or non-responder bias. Furthermore, the sophisticated methods developed for GePaRD (i) to identify pregnancy outcomes [17], which were further optimized to capture incomplete pregnancies; (ii) to link mothers' and babies' data [19]; and (iii) to estimate the beginning of pregnancy—predominantly based on the estimated date of delivery—which is expected to minimize misclassification of gestational age [18], are strengths of our study. We also consider it a strength of our study that we conducted sensitivity analyses to systematically assess whether pregnancies may have started during the recommended one-month washout period after treatment cessation.

The prevalence of isotretinoin use among girls and women of childbearing age increased in Germany between 2004 and 2019, and there were a considerable number of pregnancies likely exposed to isotretinoin in a critical period. This highlights the importance of monitoring compliance with the existing risk minimization measures for isotretinoin in Germany.

## Supporting information

**S1 Checklist. STROBE Statement—checklist of items that should be included in reports of observational studies.**
(DOCX)

**S1 Table. Number of girls and women aged 13–49 years with at least 1 dispensation of isotretinoin between 2004 and 2019 in GePaRD by age group and year of prescription.**
(DOCX)

**S2 Table. Prescriptions of Isotretinoin dispensed to girls and women aged 13–49 years between 2004 and 2019 in GePaRD: Distribution of the specialty of the prescribing physician.**
(DOCX)

**S3 Table. Number of pregnancies exposed to isotretinoin between 2004 and 2019 in GePaRD by age group and year of beginning of pregnancy.**
(DOCX)

## Acknowledgments

The authors would like to thank Marieke Niemeyer and Philipp Alexander Volkmar for statistical programming of analysis datasets and double-independent programming of results as well as all statutory health insurance providers which provided data for this study, namely AOK Bremen/Bremerhaven, DAK-Gesundheit, Die Techniker Krankenkasse (TK), and hkk Krankenkasse.

## Author Contributions

**Conceptualization:** Jonas Reinold, Nadine Wentzell, Ulrike Haug.

**Data curation:** Bianca Kollhorst.

**Formal analysis:** Bianca Kollhorst.

**Funding acquisition:** Ulrike Haug.

**Methodology:** Bianca Kollhorst, Ulrike Haug.

**Project administration:** Jonas Reinold.

**Supervision:** Ulrike Haug.

**Visualization:** Jonas Reinold.

**Writing – original draft:** Jonas Reinold.

**Writing – review & editing:** Jonas Reinold, Bianca Kollhorst, Nadine Wentzell, Katharina Platzbecker, Ulrike Haug.

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
