## [Editor Report · Decision Letter 0]

7 Oct 2022

Dear Dr Reinold, 

Thank you for submitting your manuscript entitled "Use of isotretinoin among girls and women of childbearing age and occurrence of isotretinoin-exposed pregnancies in Germany" for consideration by PLOS Medicine.

Your manuscript has now been evaluated by the PLOS Medicine editorial staff as well as by an academic editor with relevant expertise and I am writing to let you know that we would like to send your submission out for external peer review.

Please re-submit your manuscript within two working days, i.e. by Oct 11 2022 11:59PM.

Kind regards,

Philippa Dodd, MBBS MRCP PhD

Senior Editor

PLOS Medicine

---

## [Decision Letter · Decision Letter 1]

24 Aug 2023

Dear Dr. Reinold,

Thank you very much for submitting your manuscript "Use of isotretinoin among girls and women of childbearing age and occurrence of isotretinoin-exposed pregnancies in Germany" (PMEDICINE-D-22-03260R1) for consideration at PLOS Medicine. 

[LINK]

In light of these reviews, I am afraid that we will not be able to accept the manuscript for publication in the journal in its current form, but we would like to consider a revised version that addresses the reviewers' and editors' comments. Obviously we cannot make any decision about publication until we have seen the revised manuscript and your response, and we plan to seek re-review by one or more of the reviewers. 

We expect to receive your revised manuscript by Sep 14 2023 11:59PM. Please email us (plosmedicine@plos.org) if you have any questions or concerns.

We look forward to receiving your revised manuscript. 

Sincerely,

Philippa Dodd, MBBS MRCP PhD

PLOS Medicine

plosmedicine.org

COMMENTS FROM THE ACADEMIC EDITOR

My main concern with this is the lack of detail on the identification of "relevant" congenital anomalies. This is unclear and the data presented are insufficient.

Standard, rigorous, definitions of major congenital anomalies reportable to Eurocat should be used. Further details of who, and how, assessments were made, along with ICD10 coding of the anomaly and further treatments.

Figure 2 only shows the base case. Of the anomalies presented Congenital cystic liver disease is hereditary (so not influenced by exposure); Pyloric stenosis and hip dysplasia would not usually be reportable to EUORCAT (minor); and the nature and relevance of the inner ear, eye(other) and cardiac (other) anomalies are unclear. Further information about the coded diagnoses and why these have been considered "relevant" is necessary.

There is not presentation at all of the 30% of congenital anomalies referenced in the sensitivity analysis.

These figures need to be carefully described and accurately presented - I would prefer with appropriate confidence intervals around them, so as to show the imprecision of estimates given the small numbers.

As the authors point out - unintentional pregnancy while taking retinoids is a problem. Careful and accurate information about biologically plausible abnormalities should be provided, so as women can make appropriate decisions about pregnancy continuation.

At the moment the presentation feels sensationalistic- "up to 30% abnormal babies" could become a take home message, and it is not backed up by data, nor qualified with the appropriate caveats or limitations of the study. There is a risk that women could be encouraged to terminate pregnancies on the basis of this publication, assuming the risk to be much higher than it actually is. I would therefore propose major revision.

COMMENTS FROM THE EDITORS

GENERAL

Please respond to all editor and reviewer comments detailed below in full.

Please ensure that the study is reported according to the STROBE guideline, and include the completed STROBE checklist as Supporting Information. Please add the following statement, or similar, to the Methods: "This study is reported as per the Strengthening the Reporting of Observational Studies in Epidemiology (STROBE) guideline (S1 Checklist)."

When completing the checklist, please use section and paragraph numbers, rather than page or line numbers as these often change in the event of publication.

*** When reporting and discussing your results you consistently refer to percentages as opposed to whole numbers which can be un-/mis-informative. Throughout all sub-sections of the manuscript please clearly detail numerators and denominators to provide complete information to the reader ***

TITLE

Please revise your title according to PLOS Medicine's style. Your title must be nondeclarative and not a question. It should begin with main concept if possible. "Effect of" should be used only if causality can be inferred, i.e., for an RCT. Please place the study design ("A randomized controlled trial," "A retrospective study," "A modelling study," etc.) in the subtitle (ie, after a colon).

DATA AVAILABILITY STATEMENT

Thank you for your clear statement. Please define the abbreviation ‘BIPS’. Please include contact information for data requests (web or email address). Note that a study author cannot be the contact person for the data.

ABSTRACT

Please structure your abstract using the PLOS Medicine headings (Background, Methods and Findings, Conclusions).

Please combine the Methods and Findings sections into one section, “Methods and findings”.

Abstract Methods and Findings:

Please ensure that all numbers presented in the abstract are present and identical to numbers presented in the main manuscript text.

Please include the study design (i.e., observational cohort study), number of participants, length of follow up, and main outcome measures.

Please quantify the main results with 95% CIs and p values. When reporting p values, please report as p<0.001 and where higher the exact p values as p=0.002, for example. If not, to help facilitate transparent data reporting, please clearly describe the reason(s) why not.

Please include any important dependent variables that are adjusted for in the analyses.

Please include numerators and denominators used to derive percentages.

Please include a summary of adverse events assessed in the study.

In the last sentence of the Abstract Methods and Findings section, please describe the main limitation(s) of the study's methodology.

Abstract Conclusions:

Please address the study implications without overreaching what can be concluded from the data; the phrase "In this study, we observed ..." may be useful.

Please interpret the study based on the results presented in the abstract, emphasizing what is new without overstating your conclusions.

Please avoid vague statements such as "these results have major implications for policy/clinical care". Mention only specific implications substantiated by the results.

Please avoid assertions of primacy ("We report for the first time....")

AUTHOR SUMMARY

At this stage, we ask that you include a short, non-technical Author Summary of your research to make findings accessible to a wide audience that includes both scientists and non-scientists. The authors summary should consist of 2-3 succinct bullet points under each of the following headings:

• Why Was This Study Done? Authors should reflect on what was known about the topic before the research was published and why the research was needed.

• What Did the Researchers Do and Find? Authors should briefly describe the study design that was used and the study’s major findings. Do include the headline numbers from the study, such as the sample size and key findings. 

• What Do These Findings Mean? Authors should reflect on the new knowledge generated by the research and the implications for practice, research, policy, or public health. Authors should also consider how the interpretation of the study’s findings may be affected by the study limitations. In the final bullet point of ‘What Do These Findings Mean?’, please describe the main limitations of the study in non-technical language.

The Author Summary should immediately follow the Abstract in your revised manuscript. This text is subject to editorial change and should be distinct from the scientific abstract. Please see our author guidelines for more information: https://journals.plos.org/plosmedicine/s/revising-your-manuscript#loc-author-summary

INTRODUCTION

Line 23 – please replace ‘Background’ with ‘Introduction’

Line 24 – not sure that ‘application’ is the best word to use here. Perhaps instead, ‘Systemic (oral) treatment using the vitamin A derivative…is indicated in moderate to severe acne…’

Line 30 – please remove the word ‘also’

Line 49 – the use of the term ‘occurrence’ lacks nuance. Do you mean incidence or prevalence, perhaps?

Please indicate whether your study is novel and how you determined that. 

If there has been a systematic review of the evidence related to your study (or you have conducted one), please refer to and reference that review and indicate whether it supports the need for your study.

METHODS and RESULTS

Did your study have a prospective protocol or analysis plan? Please state this (either way) early in the Methods section.

For all observational studies, in the manuscript text we ask that you please indicate: 

(1) the specific hypotheses you intended to test, 

(2) the analytical methods by which you planned to test them, 

(3) the analyses you actually performed and, 

(4) when reported analyses differ from those that were planned, transparent explanations for differences that affect the reliability of the study's results. If a reported analysis was performed based on an interesting but unanticipated pattern in the data, please be clear that the analysis was data-driven.

It may be worth briefly elaborating on the healthcare set-up in Germany for those unfamiliar. Is everyone insured? How well does the database capture individuals?

Lines 72, 73 and 143 – please replace ‘prevalence [of use]’ with ‘prevalent [use of isotretinoin]’ (this is an incorrect use of a noun versus adjective). 

Line 145 onwards – please include numerators and denominators used to derive percentages.

Please define the length of follow up (eg, in mean, SD, and range).

Line 148 – you use the term ‘rates’ apparently interchangeably with prevalence. Please amend. Please ensure attention to detail when describing your measured outcomes using the same term throughout for consistency and clarity.

Please provide the actual numbers of events for the outcomes, not just summary statistics or ORs.

Please present numerators and denominators used to derive percentages.

Please quantify results with 95% CIs and p values. When reporting p values please report as p<0.001 and where higher, the exact p value as p=0.002, for example. 

When a p value is given please also report the statistical test used to determine it. If not reporting p values for the purpose of transparent data reporting please clearly state the reason(s) why not.

Please indicate whether analyses are adjusted or unadjusted and where relevant the factors that are adjusted for. Where adjusted analyses are reported, to help facilitate transparent data reporting, please also report unadjusted analyses for comparison.

TABLES

Please provide a table showing the baseline characteristics of the study population, suggest including this as table 1 and incorporating information presented in the current table 1 into it.

Please ensure that each table is affiliated to an appropriate caption which clearly describes its content without the need to refer to the text.

Please clearly indicate whether your analyses are adjusted and if so for which factors.

Where adjusted analyses are presented please also present unadjusted analyses for comparison.

FIGURES

Please ensure that each figure is affiliated to an appropriate caption which clearly describes its content without the need to refer to the text.

Please consider using a color palate suitable to those with color blindness in order to make your figures more accessible. 

DISCUSSION

Please present and organize the Discussion as follows: a short, clear summary of the article's findings; what the study adds to existing research and where and why the results may differ from previous research; strengths and limitations of the study; implications and next steps for research, clinical practice, and/or public policy; one-paragraph conclusion. We think that the main structural elements are there but additional detail could be helpful in some places – the strengths and limitations, for example.

Line 322 – please remove the funding statement and include only in the submission form – it will be compiled as metadata in the event of publication.

Line 327 – please remove the ethics statement and include only in the methods section of the your manuscript.

Line 336 – please remove the conflict of interest statement and include only in the submission form it will be compiled as metadata in the event of publication.

Line 350 – please remove the data availability statement and include only in the manuscript submission form it will be compiled as metadata in the event of publication.

REFERENCES

For in-text reference callouts citations should be placed in square brackets as follows, ‘…[1,3,6]’. Please note the absence of spaces between different citations.

In the bibliography please ensure that up to but no more then 6 author names are listed followed by et al.

Please ensure all web references include an accessed date.

Please see our website for other reference guidelines https://journals.plos.org/plosmedicine/s/submission-guidelines#loc-references

SUPPORTING INFORMATION

Please ensure to cite your Supporting Information as outlined here: https://journals.plos.org/plosmedicine/s/supporting-information

As above, please include the STROBE checklist and a copy of a relevant protocol and/or statistical analysis plan as available.

Table S1 – what do the percentages show here? Suggest placing total numbers at the top of the table for clarity. The final column could then read ‘Total study population’.

Fig S2 – is it necessary that these data be presented in both a table and a figure? It stands to reason that prescribing would be primarily by dermatologists and GPs which is what the data show. There’s nothing surprising about this, unless we missed something. Notably, I actually prefer the figure.

Comments from the reviewers:

Reviewer #1: Alex McConnachie, Statistical Review

The paper by Reinold and colleagues reports on the use of isotretinoin among women of childbearing age between 2004 and 2019 in Germany using a database covering 20% of the national population, linked to data on pregnancy and child outcomes. This review considers the statistical aspects of the paper.

On the one hand, there is not much to review. The authors present the data quite clearly, but do not use any statistical tests to analyse the data. This is fine; sometimes there is no need for statistical modelling to make the point.

However, reading the paper, I felt that there was a gap, in that the incidence of congenital malformations following isotretinoin-exposed pregnancies is reported only for the base case. Table 2 reports pregnancy outcomes for the base case plus a number of alternative scenarios as sensitivity analyses. This table could easily include a row showing the overall number and percentage of live births leading to congenital malformations. Currently, only the data for the base case is presented, in the text and in Figure 2. Adding the overall numbers for the sensitivity analyses would be useful, I think. One would hope that there are fewer malformations amongst those pregnancies less likely to be exposed to the drug. It may, however, require considerable effort to obtain this information.

My main concern with the paper, and possibly beyond my remit, is in Figure 2. This reports very low numbers of individuals. In my experience, working with unconsented data for research purposes, it is usual for reports involving small numbers of individuals to be suppressed, to prevent identification of individuals. It could be that the child with multiple malformations is unique in Germany over this time period. Someone who knows about this particular case could identify them in this paper. Is this individual woman aware that they were exposed to isotretinoin during their pregnancy? Would they find it acceptable that other people could work this out by reading the paper?

I do not know what the journal policy is on the publication of information about small numbers of individuals with rare conditions, without the consent of the patients involved, but I feel this needs to be considered.

Reviewer #2: The authors have presented a clear and well written study about the increasing use of isotretinoin in Germany and occurrence of isotretinoin exposed pregnancies. I think it is an important contribution to the literature and highlights the importance of continuing to monitor the adherence to pregnancy prevention programs. 

My main concern is how the number of exposed days was estimated. The problem with DDDs is that it is the average dose for the main indication which may not correspond to a real dose that any individual uses. For example, in Norway today, 30 tablet packages are sold with 20 mg, to be taken once or twice daily. This means that the 30 mg dose is likely not representative of any patient's real daily dose. Would it make more sense to assume 30 days-supply from the prescription date for the main analysis or as a sensitivity analysis?

Background, lines 46-47: Rather than "late treatment cessation before pregnancy," it would make more sense to phrase as: "…the extent to which conceptions occurring shortly after discontinuation of treatment contributes to the number of exposed pregnancies." It was not immediately clear what you meant. 

A similar sentence was written in the discussion, lines 215-216: "…It seems that isotretinoin is stopped too shortly before pregnancy." Again, this works the other way. Pregnancies occur to shortly after stopping treatment. This is a violation of the pregnancy prevention program and should be avoided. 

Very few pregnancies included were clearly classified as spontaneous abortions. I think this could be described as a limitation because it is unclear whether the observations of "No pregnancy outcome was recorded" are spontaneous or induced. For this reason, in the discussion, line 212: "About 45%" could be changed to "at least" 45% since there were 20.7% where the outcome was not recorded and these are assumed to be abortions. 

Methods, lines 110-111: You may wish to describe the qualifications or background of those carrying out the code profile review.

Figure S2 is not mentioned in the results section. Please add this (likely to be re-numbered S1).

Error in Table 2: % of pregnancies with "No pregnancy outcome was recorded" - for sensitivity analysis A) this should say 23.0%

Reviewer #3: In this study, the authors described the isotretinoin use in women of childbearing age in Germany, the occurrence of isotretinoin-exposed pregnancies, and malformations among children exposed in utero. I have a couple of comments for the authors to consider:

1. For the definition of exposure, why the first eight weeks of pregnancy was used, instead of other number of weeks? If two prescriptions overlap, do you add up their DDD?

2. Based on the Methods section, I am not entirely sure how the authors required health insurance coverage of the patients and the follow-up of the patients in the study. It would be great if the authors could clarify this. 

3. Based on Table 2, it seems the % of pregnancy outcomes varied substantially based on different definitions of exposure. I think this is one of the major limitations of the study. I strongly recommend the authors to expand the discussion on this in Limitations and how it impacts the interpretation of the study results. 

4. Given the small sample size, I would be careful about using the relative measures describing the changes over the years. For example, 63% increase of isotretinoin use was described while the absolute change was only 0.7 per 1000 patients from 2004 to 2019. Same for the number of isotretinoin exposed pregnancies. I suggest the authors tone down the conclusions based on relative measures and focus more on the absolute measure.

[LINK]

---

## [Decision Letter · Decision Letter 2]

7 Dec 2023

Dear Dr. Haug,

Thank you very much for re-submitting your manuscript "Use of isotretinoin among girls and women of childbearing age and occurrence of isotretinoin-exposed pregnancies in Germany: A descriptive study" (PMEDICINE-D-22-03260R2) for review by PLOS Medicine.

I have discussed the paper with my colleagues and the academic editor and it was also seen again by 2 reviewers. I am pleased to say that provided the remaining editorial and production issues are dealt with we are planning to accept the paper for publication in the journal.

[LINK]

We look forward to receiving the revised manuscript by Dec 14 2023 11:59PM.   

Best wishes,

Pippa

Philippa Dodd, MBBS MRCP PhD

PLOS Medicine

plosmedicine.org

pdodd@plos.org

COMMENTS FROM THE ACADEMIC EDITOR

The authors have done a good job of addressing comments and I am happy. One small additional from me - would the authors consider using preferred terminology of miscarriage rather than the term "spontaneous abortion".

COMMENTS FROM THE EDITORS

GENERAL

Thank you for your detailed and considered responses to previous editor and reviewer comments. Please see below for further comments which we require that you address prior to publication.

We agree with Academic Editor (above) that revising your use of terminology would be preferable. Please amend throughout all subsections of the manuscript and supporting files.

TITLE

Please revise the title replacing ‘descriptive’ with ‘population-based’.

ABSTRACT

Line 10 – ‘prescription’ might be preferable to ‘supply’ for explicit clarity.

Line 15/16 – please clearly define the numerical values contained within brackets (confidence intervals). Please use commas instead of hyphens to separate upper and lower bounds as the latter can be confused with reporting of negative values. We accept your response regarding inclusion of p values.

AUTHOR SUMMARY

Line 43 – suggest ‘…period, it is possible that additional pregnancies could have also been exposed when the pregnancy was conceived before the end of the washout period.’ Or similar.

Line 47 – suggest ‘were’ instead of ‘was’.

Line 54 – suggest ‘…dose of isotretinoin’.

METHODS and RESULTS

As above please amend statistical reporting to separate upper and lower CI bounds with commas as opposed to hyphens. Please check and amend throughout all sub-sections of the manuscript and supporting files.

Line 234 – please replace ‘N’ with ‘n’ for consistency.

FIGURES

Figure 1 – please define GePaRD in the caption. Please indicate in the caption that the shaded areas represent 95% confidence intervals (not just on the axis).

DISCUSSION 

We appreciate your inclusion of sub-headings to direct the reader to the different parts o your study question. We generally prefer to format the discussion as continuous prose and suggest that actually sub-headings are not necessary. If you disagree, then a compromise could be leaving in those at lines 297 and 320 and removing the rest.

Line 285 – suggest stating ‘20% of the population’ as per elsewhere in the manuscript.

Line 286 – ‘increased by 63%’ is correct in proportional terms but somewhat overinflates the magnitude of the issue when considering the data that follows. Please revise so as not to overstate findings.

Line 289 – please quantify the number of users that you derive this percentage from.

Line 291 – suggest ‘…most critical for fetal development.’ instead perhaps.

Line 294 – is this 45% of 178? It might be helpful to simply state the number here as opposed to the percentage considering the discussion of outcomes also from your sensitivity analyses which could confuse matters.

Line 421 – please remove the funding statement from the end of the manuscript and include only in the manuscript submission form when you re-submit the masncuript. It will be compiled as metadata at the time of publication.

We appreciate, as you say that the study was not designed to assess level of risk, but you do quote numbers of congenital abnormalities. As such we think it would be helpful to report the expected risk of congenital abnormalities and atrial septal defects, in the German population, at least in the discussion (and perhaps also in the abstract).

REFERENCES

Line 444 – ref #5 please remove the word ‘date’ and just detail the actual date. Please check and amend elsewhere as necessary (I note also refs 12 and 15 but this is not exhaustive).

SUPPORTING INFORMATION

In the published article, supporting information files are accessed only through a hyperlink attached to the captions. For this reason, you must list captions at the end of your manuscript file. You may include a caption within the supporting information file itself, as long as that caption is also provided in the manuscript file. Do not submit a separate caption file.

Throughout, please define GePaRD

S1 checklist – thank you for including the checklist as required. Please change column header ‘Page No’ to ‘Section & Paragraph’ or similar. Please also revise to indicate that your statements (funding, for example) can be found in the submission form as metadata.

STATEMENTS FOR SUBMISSION

As detailed above please ensure all are removed from the main manuscript and included only in the manuscript submission form when you re-submit the manuscript.

SOCIAL MEDIA

To help us extend the reach of your research, please detail any X (formerly Twitter) handles you wish to be included when we tweet this paper (including your own, your coauthors’, your institution, funder, or lab) in the manuscript submission form when you re-submit the manuscript.

COMMENTS FROM THE REVIEWERS

Reviewer #1: Alex McConnachie, Statistical Review

I thank the authors for their consideration of my original points. I am very glad that they went to the extra effort to obtain the outcomes for children classified as exposed in the sensitivity analyses - I think this makes the data more complete. I am also satisfied with the additional confirmation that it is ethically acceptable to publish these data in such detail.

I have no further comments.

Reviewer #2: Thank you for the clear response. You have addressed all of my comments. Two minor things: for consistency, I recommend to add "at least" before "45% of exposed pregnancies ended in an induced abortion" in the abstract. In the revision on page 18, line 333 there is a typo: :"to" should be "too"

[LINK]

---

## [Editor Report · Decision Letter 3]

21 Dec 2023

Dear Dr Haug, 

On behalf of my colleagues and the Academic Editor, Dr Sarah Stock, I am pleased to inform you that we have agreed to publish your manuscript "Use of isotretinoin among girls and women of childbearing age and occurrence of isotretinoin-exposed pregnancies in Germany: A population-based study" (PMEDICINE-D-22-03260R3) in PLOS Medicine.

PRESS

Thank you again for submitting to PLOS Medicine, it has been a pleasure handling your manuscript. We look forward to publishing your paper. 

Best wishes,

Pippa 

Philippa Dodd, MBBS MRCP PhD

PLOS Medicine

pdodd@plos.org